# Interplay between kinesin-1 and cortical dynein during axonal outgrowth and microtubule organization in *Drosophila* neurons

Urko del Castillo, Michael Winding, Wen Lu, Vladimir I Gelfand*

Department of Cell and Molecular Biology, Feinberg School of Medicine, Northwestern University, Chicago, United States

**Abstract** In this study, we investigated how microtubule motors organize microtubules in *Drosophila* neurons. We showed that, during the initial stages of axon outgrowth, microtubules display mixed polarity and minus-end-out microtubules push the tip of the axon, consistent with kinesin-1 driving outgrowth by sliding antiparallel microtubules. At later stages, the microtubule orientation in the axon switches from mixed to uniform polarity with plus-end-out. Dynein knockdown prevents this rearrangement and results in microtubules of mixed orientation in axons and accumulation of microtubule minus-ends at axon tips. Microtubule reorganization requires recruitment of dynein to the actin cortex, as actin depolymerization phenocopies dynein depletion, and direct recruitment of dynein to the membrane bypasses the actin requirement. Our results show that cortical dynein slides 'minus-end-out' microtubules from the axon, generating uniform microtubule arrays. We speculate that differences in microtubule orientation between axons and dendrites could be dictated by differential activity of cortical dynein.

**\*For correspondence:** vgelfand@northwestern.edu

**Competing interests:** The authors declare that no competing interests exist.

## Introduction

Neuronal development involves the dramatic morphologic reorganization of spherical morphologically undifferentiated cells to highly polarized mature neurons. Developing neurons grow long microtubule-based axons and dendrites. Over the last decades, the origin of the mechanical forces that drive process formation in neurons has been the subject of numerous studies (*Suter and Miller, 2011*; *Kapitein and Hoogenraad, 2015*). Recently, our lab demonstrated that kinesin-1, a major microtubule motor, slides microtubules against each other in many cell types (*Jolly et al., 2010*; *Barlan et al., 2013*), including neurons, and this sliding plays an important role in neuronal polarization. We have shown that microtubule-microtubule sliding is required for initial axon formation and regeneration after injury (*Lu et al., 2013b*; *2015*). This process is likely required for neurodevelopment in many organisms besides *Drosophila*, as kinesin-driven microtubule sliding has been implicated in generating the minus-end-out microtubule pattern observed in dendrites of *Caenorhabditis elegans* neurons (*Yan et al., 2013*).

The main well-established function of kinesin-1 (also known as conventional kinesin) is the transport of cargoes along microtubules in the cytoplasm. Each kinesin-1 molecule is a heterotetramer that consists of two heavy chains (KHC) and two light chains (*Kuznetsov et al., 1988*). Each KHC polypeptide contains two microtubule-binding domains: one ATP-dependent site in the motor domain and a second ATP-independent site at the C-terminus (*Hackney and Stock, 2000*; *Seeger and Rice, 2010*; *Yan et al., 2013*). Kinesin-1 is thought to slide microtubules against each other with these two heavy chain domains; one microtubule is used as a track, while the other is

**eLife digest** Motor proteins can move along filaments called microtubules to transport proteins and other materials to different parts of the cell. Microtubules are "polar" filaments, meaning that they have two distinct ends that have different chemical properties. Motor proteins can only move along these filaments in one direction, for example, the kinesin motor proteins generally move toward the so-called "plus-end", while dynein motors move in the opposite direction.

A typical nerve cell (or neuron) is composed of a cell body, a long projection called an axon and many small branch-like structures called dendrites. Within the axon, the microtubules are arranged so that their plus-ends point outwards, but the microtubules in dendrites are arranged differently so that many minus-ends point outwards instead. This polarity is important for the neuron in deciding which proteins should be transported to axons, and which should go to the dendrites. However, it is not clear how these different microtubule arrangements are established.

Here, del Castillo et al. used microscopy to study microtubules in the axons of fruit fly neurons. The experiments show that in the very early stages of neuron development, the axons contained microtubules of mixed polarity. However, by the later stages, the microtubules had become uniform with all the plus-ends directed outwards.

Further experiments show that dynein is responsible for this organization as it pushes the minus-end-out microtubules out of the axons. Dynein uses a scaffold made of a protein called actin to attach to the inner surface of the cell and move the minus-end microtubules to the cell body of the neuron. Thus, del Castillo et al.'s findings reveal that these dynein motors are responsible for the polarity of microtubules in mature axons. The next challenge is to understand how dynein is attached to the actin scaffold and why it rearranges microtubules in axons, but not in dendrites.

transported as a cargo; kinesin light chains are not required for sliding (*Jolly et al., 2010*; *Yan et al., 2013*).

Axons contain microtubule arrays of uniform orientation with plus-ends facing the axon tip (*Baas et al., 1988*; *Stone et al., 2008*). However, kinesin-1 is a plus-end motor, and therefore can only slide microtubules with their minus-ends leading and plus-ends trailing (*Figure 1A*), which is inconsistent with the final orientation of microtubules in mature axons. To address this apparent contradiction, we asked two questions: First, are microtubules indeed pushed with their minus-ends out at the initial stages of axon outgrowth, as would be expected if they are pushed by kinesin-1? Second, if this is the case, how are microtubules with the 'wrong' orientation replaced by microtubules with normal (plus-end-out) orientation in mature axons? To address these questions, we imaged and tracked markers of microtubule plus-ends and minus-ends in cultured *Drosophila* neurons and S2 cells at different stages of process growth. Our results showed that, at the initial stages of neurite formation, microtubules have mixed polarity with minus-ends being pushed against the plasma membrane; later, cytoplasmic dynein, attached to the actin cortex, removes minus-end-out microtubules to the cell body, creating microtubule arrays with uniform plus-end-out orientation. We speculate that regulation of dynein's microtubule sorting activity could explain the differences in microtubule orientation between axons and dendrites.

## Results

### Microtubule minus-ends push neurite tips at the initial stages of process formation

We previously demonstrated that kinesin-1 slides microtubules against each other, and this sliding generates the forces that drives outgrowth at the initial stages of neurite outgrowth (*Lu et al., 2013b*) and axon regeneration (*Lu et al., 2015*). Because kinesin-1 is a plus-end microtubule motor, it can only slide microtubules with their minus-ends leading and plus-ends trailing (*Figure 1A*). If this model is correct, it suggests that kinesin-1 must extend neurites by pushing microtubule minus-ends against the plasma membrane during the initial stages of neurite formation. Furthermore, because the model predicts that two microtubules have to be in antiparallel orientation to slide against each

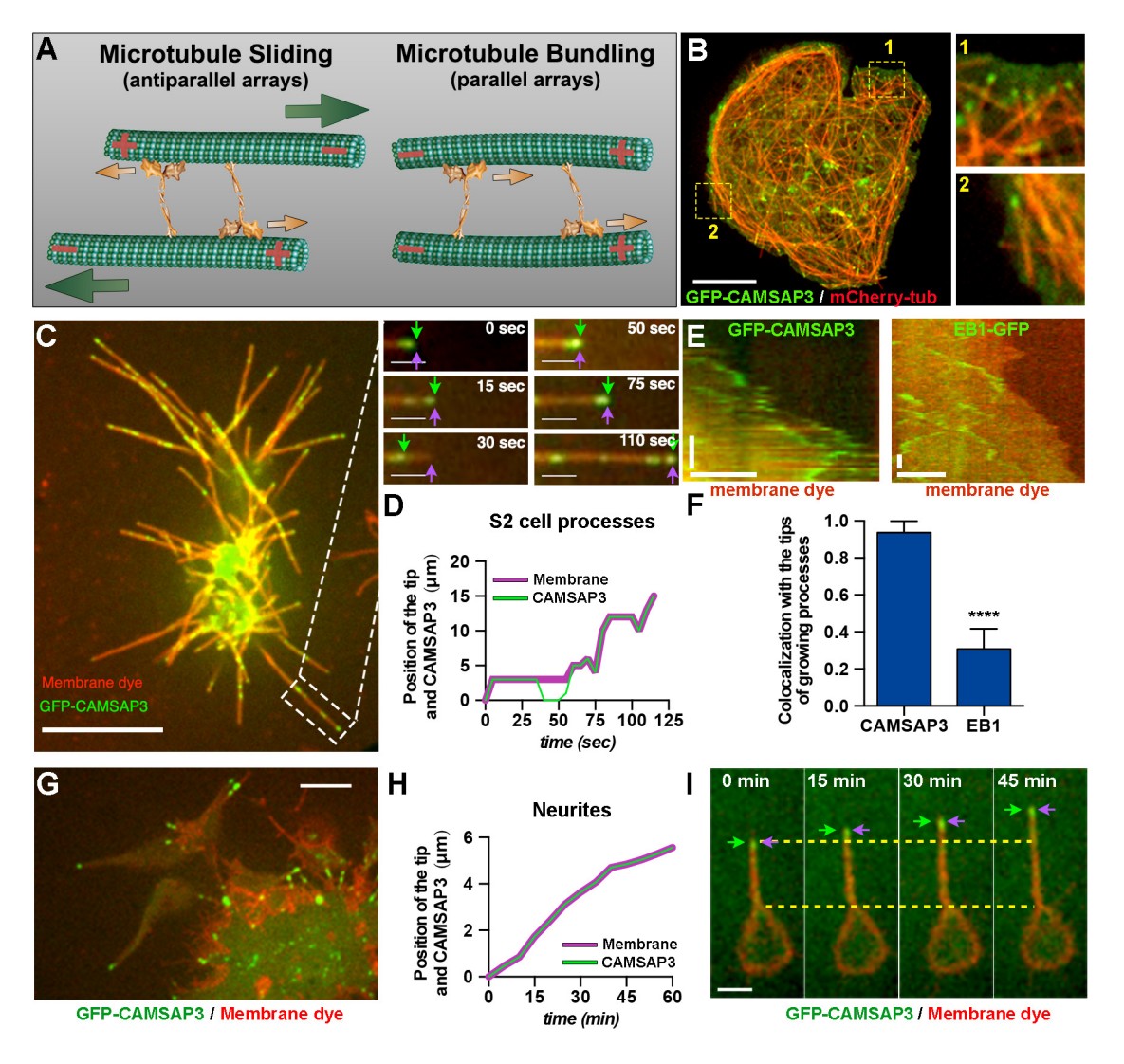

**Figure 1.** Microtubule minus-ends push the plasma membrane during the initial stages of neurite outgrowth. (**A**) Model of microtubule-microtubule sliding driven by kinesin-1. Kinesin-1 slides antiparallel microtubules apart with their minus-ends leading (left panel). When kinesin-1 binds to parallel microtubules (right panel), forces applied by the two motors to the two microtubules are counteracted resulting in no net movement; instead, kinesin-1 crosslinks these microtubules. Large green arrows indicate direction of microtubule sliding; small orange arrows indicate the direction of kinesin-1 movement relative to microtubules. (**B**) A representative S2 cell expressing GFP-CAMSAP3 and mCherry-tubulin. Note that CAMSAP3 molecules accumulate at microtubule ends. Two different regions of the cell body (labeled 1 and 2) were magnified in the insets (see *Video 2*). Scale bar, 5 μm. (**C** and **D**) Minus-ends of microtubules localize at the tips of growing processes during the initial stages of process formation in S2 cells. GFP-CAMSAP3 expressing S2 cells were plated on coverslips and imaged 5 min after plating. The plasma membrane was stained with a Deep Red membrane dye (red). (**C**) Last frame of a time-lapse video. Images at different time points of the growing process in the white box are shown at higher magnification. Green arrows indicate positions of the most distal CAMSAP3 dot; magenta arrows show the position of the tip of the process (see *Video 4*). Scale bars are 10 μm and 3 μm for main and inset panels, respectively. (**D**) A graph showing the position of the process tip and the microtubule minus-ends shown in the inset of (C) as a function of time. (**E–F**) Microtubule plus-ends do not colocalize with the tip of growing processes in S2 cells. (**E**) Representative kymographs of growing processes from cells expressing GFP-CAMSAP3 (left panel) or EB1-GFP (right panel). The plasma membrane was stained with a Deep Red membrane dye. Note that CAMSAP3 consistently localizes at the tips of the processes during outgrowth events, however EB1 comets do not colocalize with the tip of the growing processes (horizontal scale bar, 10 μm; vertical scale bar, 25 s). (**F**) Graph depicting the fraction of time that CAMSAP3 or EB1 colocalize with the tips of the processes during the growing events. Error bars indicate s.d. (CAMSAP3, n=55 growing processes; EB1, n=51 growing processes). Data collected from four independent experiments. ****p<0.0001. (**G–I**) Localization of microtubule minus-ends at the tips of the processes during the initial stages of neurite formation in cultured neurons. (**G**) A still image of 4 hr-cultured neurons expressing *elav>GFP-CAMSAP3*. The plasma membrane was labeled with Deep Red dye. Note that CAMSAP3 mostly localized to the tips of neurites. Scale bar, 5 μm. (**H**) Diagram showing the position of the neurite tip and the microtubule minus-ends of the axon showed in (I) as a function of time. (**I**) Still images from a

*Figure 1 continued on next page*

*Figure 1 continued*

time-lapse of a 4 hr-cultured neuron plated as described in (**G**). Yellow dashed lines are guides to visualize the neurite growth. Green arrows indicate positions of CAMSAP3; magenta arrows show position of the tip of the process (see **Video 5**). Scale bar, 5 µm.

The following figure supplement is available for figure 1:

**Figure supplement 1.** CAMSAP3 labels minus-ends of microtubules in *Drosophila* S2 cells.

other, sliding by kinesin-1 will result in the simultaneous transport of two microtubules in opposite directions (see *Figure 1A* and the legend for the explanation). Bidirectional microtubule movement can indeed be observed in growing axons of cultured *Drosophila* neurons using tubulin tagged with a photoconvertible probe (*Video 1*).

To initially test this hypothesis, we first took advantage of *Drosophila* S2 tissue culture cells. S2 cells provide a good model system to explore the mechanism of process formation because they canbe induced to form cellular processes when the integrity of the actin filament network is impaired by treatment with either Cytochalasin D or Latrunculin B (LatB) (*Kim et al., 2007*; *Lu et al., 2013a*). In addition, this system enables us to efficiently study the mechanisms of process formation by knocking down candidate proteins with double-stranded RNA (dsRNA) (*Rogers and Rogers, 2008*).

To study microtubule minus-end distribution in live cells, we ectopically expressed a fluorescently tagged microtubule minus-end binding protein called calmodulin-regulated spectrin-associated protein (CAMSAP), also known as Patronin or Nezha. CAMSAP proteins bind to microtubule minus-ends and stabilize them against depolymerization, making them the perfect candidate to label microtubule minus-ends (*Akhmanova and Hoogenraad, 2015*). We initially performed experiments with GFP-tagged Patronin, the single *Drosophila* member of CAMSAP family (*Wang et al., 2013*), but its expression level in S2 cells was very low and GFP signal was not robustly found on microtubules (data no shown). On the other hand, its mammalian ortholog CAMSAP3 tagged with GFP expressed at consistently higher levels and reliably decorated microtubule ends (*Figure 1B*).

First, we wanted to test whether GFP-CAMSAP3 decorates only one end of microtubules in *Drosophila* cells. Because the microtubule network is normally too dense to identify both ends of microtubules, we induced the formation of short microtubules in cells by partial depolymerization with 25 µM Vinblastine for 1 hr. Examination of these short microtubule fragments demonstrated that only one end of each microtubule contained a GFP-CAMSAP3 patch (*Figure 1—figure supplement 1A*). In untreated S2 cells, spontaneous growth and shrinkage events associated with the dynamic instability of microtubule plus-ends were not seen in microtubule ends decorated by GFP-

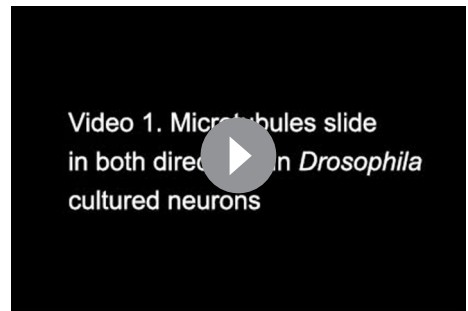

**Video 1.** Microtubules slide in both directions in *Drosophila*-cultured neurons. Time-lapse video of photoconverted microtubules in *Drosophila*-cultured neurons expressing tdEOS-αtubulin. A small area of the nascent axon was photoconverted by 405 nm light. Note that microtubules slide in both directions. Scale bar 5 µm.

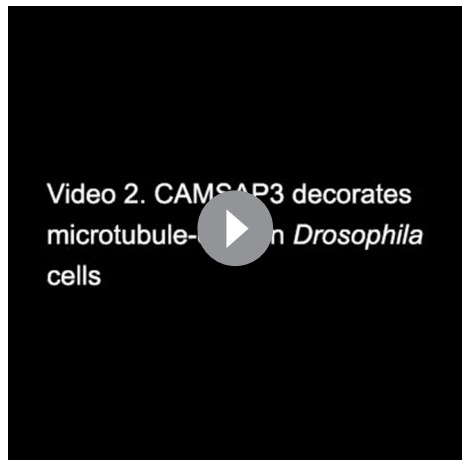

**Video 2.** CAMSAP3 decorates microtubule-ends in *Drosophila* cells. Related to *Figure 1B*. A time-lapse video of S2 cells expressing GFP-CAMSAP3 and mCherry-tubulin. Scale bar 10 µm.

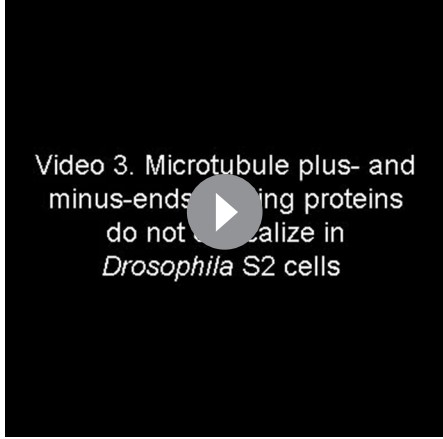

**Video 3.** Microtubule plus-ends and minus-ends binding proteins do not colocalize in *Drosophila* S2 cells. Related to *Figure 1—figure supplement 1B,C*. A time-lapse video of a S2 cell coexpressing EB1-GFP and mCherry-CAMSAP3. Scale bar, 10 μm.

**Video 4.** Initial process outgrowth in *Drosophila* S2 cells is driven by microtubule minus-ends. Related to *Figure 1C*. A time-lapse video of *Drosophila* S2 cells expressing GFP-CAMSAP3 plated for 5 min. Deep red dye was used to stain the membrane. Scale bars 10 μm and 5 μm, respectively.

CAMSAP3, suggesting that GFP-CAMSAP3 labels microtubule minus-ends in *Drosophila* cells (*Video 2*). Furthermore, EB1-GFP and mCherry-CAMSAP3 never colocalized when expressed in the same cell, further confirming the minus-end localization of CAMSAP3 (*Figure 1—figure supplement 1B and 1C*; *Video 3*). These results, together with published data (*Tanaka et al., 2012*; *Hendershott and Vale, 2014*; *Jiang et al., 2014*; *Akhmanova and Hoogenraad, 2015*), demonstrated that mammalian GFP-CAMSAP3 reliably marks microtubule minus-ends in *Drosophila*.

To study localization of microtubule minus-ends in growing processes, we induced the formation of processes in S2 cells expressing GFP-CAMSAP3 and started collecting images 5 min after plating the cells. At this time point, nascent processes were actively growing. We simultaneously tracked GFP-CAMSAP3 and the plasma membrane using a membrane dye (CellMask Deep Red). We found that a significant fraction of growing processes contained GFP-CAMSAP3 dots at their tips, and that these processes only elongated when microtubule minus-ends were present at their tips (*Figure 1C, D*; *Video 4*). Interestingly, we often observed retraction of the GFP-CAMSAP3 marker from the process tip; these events always coincided with a pause in process outgrowth (*Figure 1C*, inset). Quantitative analysis demonstrated that while CAMSAP3 almost always colocalized with the tips of the growing processes, the plus-ends marker EB1 could only be found in the tips of the growing processes approximately 30% of the time (*Figure 1E,F*), suggesting that at this stage microtubule dynamics does not play a major role in process outgrowth.

To investigate the localization of microtubule minus-ends in *Drosophila* neurons, we created a transgenic fly that expresses GFP-CAMSAP3 under the *UAS* promoter. Neurons were harvested and cultured from the brains of larvae expressing GFP-CAMSAP3 driven by the pan-neuronal promoter *elav-Gal4* (*Egger et al., 2013*; *Lu et al., 2015*) (see Material and methods). We visualized microtubule minus-ends in growing

**Video 5.** Microtubule minus-ends push the plasma membrane in growing neurites of young cultured *Drosophila* neurons. Related to *Figure 1I*. Time-lapse video of a *Drosophila* neuron expressing *elav>GFP-CAMSAP3* cultured for 4 hr. Deep red dye was used to stain the membrane. Scale bar 5 μm.

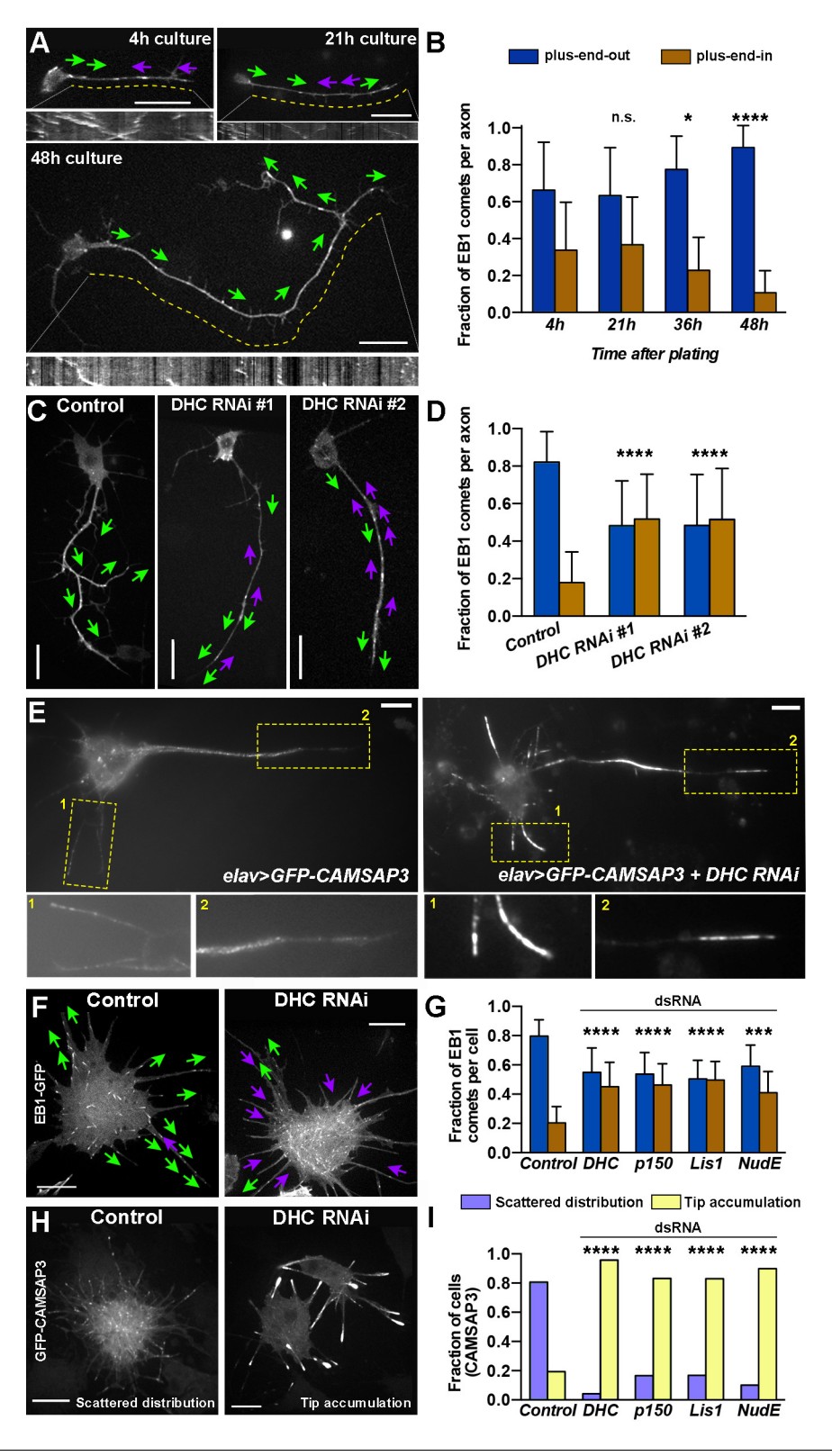

**Figure 2.** Dynein specifies microtubule orientation in axons. (**A** and **B**) Axonal microtubules gradually acquire uniform orientation during development. (**A**) Representative still images of EB1-GFP expressing neurons cultured for 4 hr, 21 hr, or 48 hr. Kymographs of EB1 comets are shown below corresponding images. Magenta and green arrows indicate direction of EB1 comet movement (plus-end-in and plus-end-out, respectively). Dashed yellow lines define the area of the axon used for plotting EB1-GFP kymographs (see *Video 6*). Scale bars, 10 μm. (**B**) Fraction of EB1-GFP comets

*Figure 2 continued on next page*

*Figure 2 continued*

directed toward the tip of the neurites (plus-end-out) or the cell body (plus-end-in). See Material and methods for an explanation of EB1 comet quantification. Error bars indicate s.d. (4 hr, n=35 axons with 761 comets; 21 hr, n=33 axons with 408 comets; 36 hr, n=33 axons with 526 comets; 48 hr, n=25 axons with 299 comets). *p=0.034, ****p<0.0001, n.s. = not significant. Data collected from three independent experiments. (C and D) Dynein knockdown causes mixed orientation of microtubules in axons. (C) Representative still images of control (*elav-gal4*) and dynein knockdown (two different DHC shRNAs driven by *elav-Gal4*) 48 hr-cultured neurons expressing EB1-GFP. Magenta and green arrows indicate directions of EB1 comet movement (plus-end-in and plus-end-out, respectively) (see *Video 7*). Scale bars, 10 μm. (D) Fraction of EB1-GFP comets directed toward the tip of the neurites (plus-end-out) or the cell body (plus-end-in). Error bars indicate s.d. (Control, n=28 axons with 269 comets; DHC RNAi#1, n=45 axons with 928 comets; DHC RNAi#2, n=30 axons with 454 comets). ****p<0.0001. Data collected from three independent experiments. (E) Dynein knockdown in neurons results in accumulation of microtubule minus-ends at the tips of neurites. Images of 48 hr-cultured neurons expressing *elav>GFP-CAMSAP3* (left panel) or *elav>GFP-CAMSAP3 + DHC RNAi* (right panel). Bottom panels are magnifications of the yellow-boxed areas. Scale bars, 10 μm. (F–G) Dynein inactivation induces antiparallel microtubule arrays in S2 cell processes. (F) Representative images of untreated (control) or DHC dsRNA-treated S2 cells expressing EB1-GFP. Magenta and green arrows indicate plus-end-in or plus-end-out direction of EB1 comet movement, respectively (see *Video 8*). Scale bars, 10 μm. (G) Graphs depict the direction of EB1-GFP comets in the processes of control S2 cells, and cells after knockdown of DHC, p150$^{Glued}$, Lis1, or NudE. Error bars indicate s.d. (Control, n=55 cells with 1747 comets; DHC RNAi, n=50 cells with 1929 comets; p150 RNAi, n=26 cells with 2282 comets; Lis1 RNAi, n=33 cells with 3359 comets; NudE RNAi, n=24 cells with 2518 comets). ***p=0.001–0.0001, ****p<0.0001. Data collected from three independent experiments. (H–I) Dynein inactivation in S2 cells results in accumulation of microtubule minus-ends in the process tips. (H) Representative images of untreated (control) or DHC dsRNA-treated S2 cells expressing GFP-CAMSAP3. In control S2 cells, CAMSAP3 particles display a scattered distribution with few minus-ends at process tips. In dynein RNAi S2 cells, CAMSAP3 particles robustly accumulate at the tips of the processes. Scales bars, 10 μm. (I) Graphs show the fraction of S2 cells displaying the phenotypes depicted in (H). (Control, n=117 cells; DHC RNAi, n=84 cells; p150$^{Glued}$=90 cells; Lis1 RNA1, n=83 cells; NudE RNAi=79 cells). DHC, dynein heavy chain. ****p<0.0001. Data collected from three independent experiments.

The following figure supplements are available for figure 2:

**Figure supplement 1.** Knockdown efficiency of DHC and dynein cofactors in *Drosophila* S2 cells.

**Figure supplement 2.** Distribution of GFP-CAMSAP3 in DHC RNAi S2 processes.

neurites at the initial stages of growth (4 hr after plating), when neurons started to develop processes (length= 9.96 μm, s.d. ± 4.5 μm, n=50 axons). We found that, like in S2 cells, the growing neurites contained GFP-CAMSAP3 dots at the tips of neurites, and localization of the dots to the tips precisely correlated with neurite outgrowth (*Figure 1G–I*; *Video 5*). All together, these data show that at least a fraction of microtubules in growing neurites have the 'wrong' orientation (minus-end-out). Localization of microtubule minus-end(s) at the neurite tip correlates with neurite outgrowth, consistent with kinesin-1 pushing the minus-ends of microtubules against the plasma membrane driving the initial outgrowth.

## Developing axons contain microtubule arrays with mixed orientation

If kinesin-1 slides antiparallel microtubules at the initial stages of axon formation, the growing neurites should initially contain microtubules with mixed orientation (*Figure 1A* and the legend for the explanation). To test this prediction, we imaged and tracked the direction of the plus-end microtubule marker, EB1-GFP, in the axons of cultured neurons at different time points after plating. Quantitative analysis of EB1-GFP comets using tracking software (see Material and methods) demonstrated that, shortly after plating, growing neurites contained EB1 comets moving in both anterograde and retrograde directions (*Figure 2A,B*; *Video 6*). The microtubule orientation in axons remained mixed during the 1st day in culture. At 36 hr, the fraction of retrograde EB1 comets started to decline, and at 48 hr, developed axons were mostly filled with plus-end-out microtubules (*Figure 2A,B*; *Video 6*). These results demonstrated that axons initially contain microtubule arrays with mixed orientation.

## Dynein sorts microtubules in *Drosophila* axons

Our data suggest that some sorting mechanism is responsible for the elimination of microtubules with the 'wrong' (plus-ends toward the soma) orientation from maturing axons. We hypothesized that this sorting factor is cytoplasmic dynein because it has been reported that mutations in dynein light intermediate chain or dynein-cofactors (NudE) are required for uniform microtubule orientation in axons of *Drosophila* dendritic arborization (da) neurons (*Zheng et al., 2008*; *Arthur et al., 2015*).

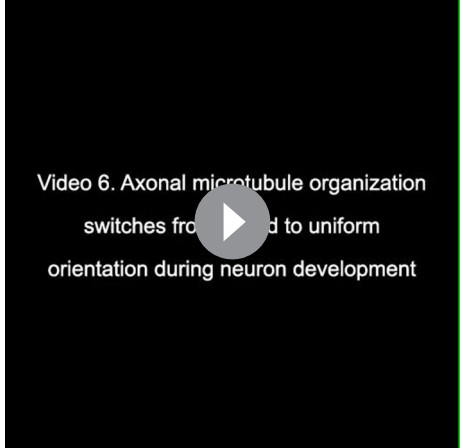

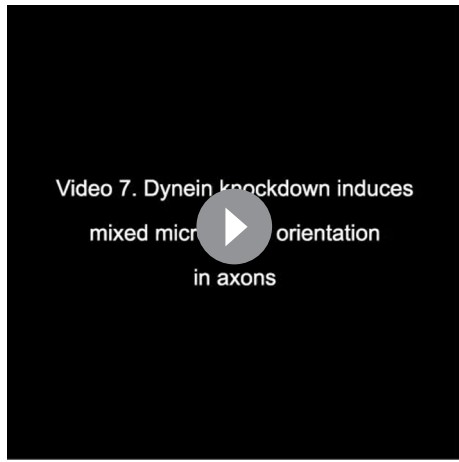

**Video 6.** Axonal microtubule organization switches from mixed to uniform orientation during neuron development. Related to *Figure 2A*. Time-lapse videos of *Drosophila*-cultured neurons expressing *ubi-EB1-GFP* cultured for 4 hr, 21 hr and 48 hr. Scale bars, 10 µm.

**Video 7.** Dynein knockdown causes axons to contain antiparallel microtubules. Related to *Figure 2C*. Time-lapse videos of a control and two *elav>DHC shRNA Drosophila* cultured neurons expressing EB1-GFP. Scale bars, 10 µm.

To characterize the role of dynein, we expressed two different shRNAs targeting dynein heavy chain (DHC) using *elav-Gal4* to specifically knock down dynein in neurons. *elav>DHC RNAi* animals developed to the third instar larval stage; however, their locomotion was severely impaired and most died before reaching the pupae stage (data not shown). In addition, none of the surviving *elav>DHC RNAi* pupae eclosed into adults. We cultured neurons obtained from brains of third instar *elav>DHC RNAi* larvae. Our lab has previously shown that mitochondrial movement was substantially diminished in *elav>DHC RNAi* neurons, indicating that the activity of dynein is impaired in those neurons (*Lu et al., 2015*).

To track the microtubule orientation after dynein depletion, we genetically combined transgenes encoding *DHC RNAi* with *EB1-GFP* or *GFP-CAMSAP3*. We first quantified the direction of the EB1 comets in neurons grown for 48 hr; at this time point, microtubules in the axons of control neurons are mostly oriented with plus-end-out (*Figure 2B*). Analysis of EB1 comets in *elav>DHC RNAi* 48 hr-neurons revealed that axons contained microtubule arrays of mixed orientation (*Figure 2C,D*; *Video 7*), suggesting that dynein is necessary to remove minus-end-out microtubules from axons. Interestingly, while microtubule minus-ends in control neurons, as visualized by GFP-CAMSAP3, were scattered throughout the length of the axons, inactivation of dynein resulted in dramatic accumulation of the minus-end markers at neurite tips (*Figure 2E*; control, left panels; *DHC-RNAi*, right panels).

To further investigate the role of dynein in organizing microtubule arrays in cell processes, we again examined *Drosophila* S2 cells. The treatment of S2 cells with DHC dsRNA efficiently knocked down DHC (*Figure 2—figure supplement 1A,B*). Recapitulating the neuronal phenotype, dynein depletion in S2 cells resulted in processes with EB1-GFP comets traveling in both directions (*Figure 2F,G*; *Video 8*). We also examined the distribution of the GFP-CAMSAP3 in S2

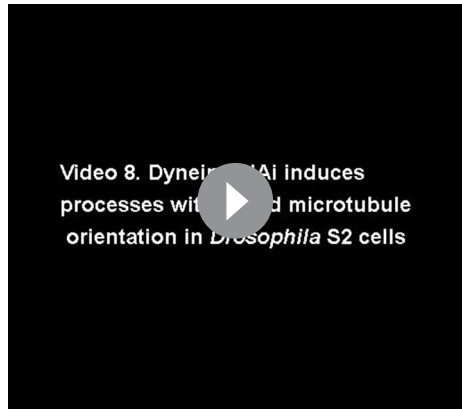

**Video 8.** Dynein RNAi causes mixed microtubule orientation in processes of *Drosophila* S2 cells. Related to *Figure 2F*. Time-lapse videos of a control (untreated) and DHC RNAi *Drosophila* S2 cells expressing EB1-GFP. Scale bars, 10 µm.

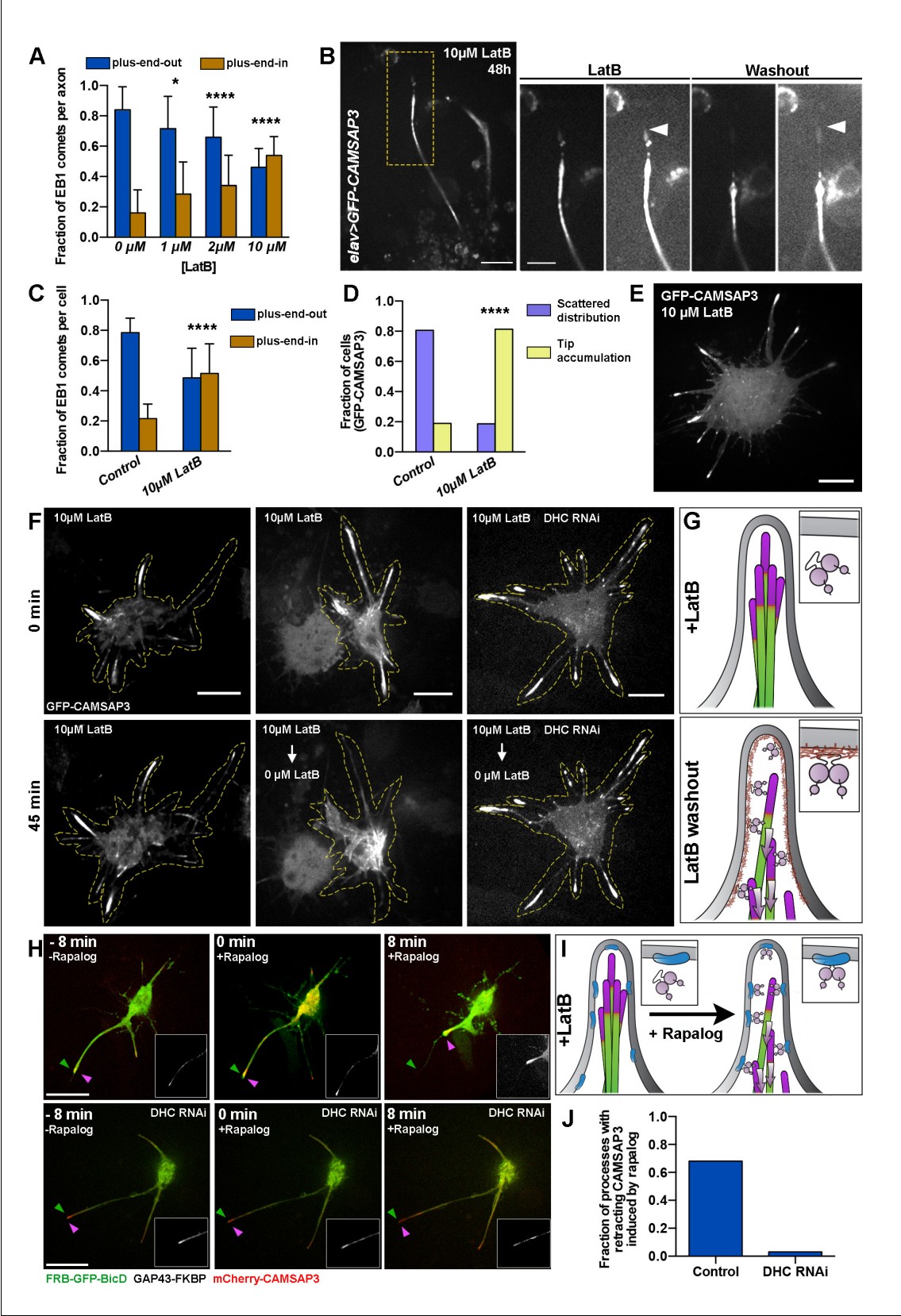

**Figure 3.** Dynein recruitment to the cortex is required for microtubule sorting. (**A**) Actin depolymerization in cultured neurons results in formation of axons with antiparallel microtubules. Graph depicts the fraction of axonal EB1 comets moving in each direction in 48 hr-cultured neurons treated with *Figure 3 continued on next page*

Figure 3 continued

different concentrations of LatB. Error bars indicate s.d. (0 µM LatB, n=34 axons with 905 comets; 1 µM LatB, n=20 axons with 852 comets; 2 µM LatB, n=37 axons with 102 comets; 10 µM LatB, n=15 axons with 547 comets). *p=0.0144, ****p<0.0001. Data collected from three independent experiments. (B) LatB treatment induces accumulation of minus-ends in neurite tips. A 48 hr-cultured neuron expressing *elav>GFP-CAMSAP3*. Panels on the right are magnified areas of the neurite tip before or 14 hr after LatB washout. Images are overexposed to show the outline of the neurite. Note that LatB washout induces retrograde movement of CAMSAP3 decorated microtubules. Scale bars are 10 µm and 5 µm, respectively. (C–E) Actin depolymerization in S2 cells results in the formation of processes with antiparallel microtubule orientation and an accumulation of microtubule minus-ends at the tips of processes (compare with *Figure 2F–H*). (C) Graph depicting the direction of EB1-GFP comets in S2 processes. Error bars indicate s.d. (Control, n=20 cells with 1833 comets; 10 µM LatB, n=22 cells with 2408 comets). ****p<0.0001. Data collected from three independent experiments. (D) Graph depicting the distribution of GFP-CAMSAP3 in the processes of S2 cells treated with 0.5 µM (control) or 10 µM LatB (Control, n=82 cells; 10 µM LatB, n=75 cells). ****p<0.0001. Data collected from three independent experiments. (E) Confocal image of a S2 cell expressing GFP-CAMSAP3 plated in 10 µM LatB. Note that GFP-CAMSAP3 accumulates at process tips. Scale bar, 10 µm. (F and G) Recruitment of dynein to cortical actin activates the sorting activity of dynein. (F) Representative still images from time-lapses of S2 cells expressing GFP-CAMSAP3 plated for 4 hr in the presence of 10 µM of LatB. In control cells, the microtubule minus-ends remain at the tips of the processes (left panels). LatB washout resulted in a clearing of microtubules minus-ends from processes (middle panels). Dynein knockdown impairs the microtubule sorting activity after LatB washout and microtubule minus-ends remained clustered at the tips (see *Video 9*). Scale bars, 10 µm. (G) Schematic representation of the LatB washout assays (microtubule minus-ends are represented in magenta). In the presence of LatB, dynein is decoupled from the plasma membrane, preventing its sorting activity (top panel). After washout, dynein is recruited to the plasma membrane by cortical actin, resulting in robust microtubule sorting and transport of minus-end-out microtubules toward the cell body (bottom panel).(H–I) Direct recruitment of dynein to the membrane bypasses the F-actin requirement for microtubule sorting. Endogenous dynein can be recruited to the plasma membrane in S2 cells coexpressing FRB-GFP-BicD and GAP43-FKBP. In the presence of 10 µM LatB, cortical actin is depolymerized and therefore dynein remains soluble in the cytoplasm. Addition of 1 µM rapalog induces the direct recruitment of the dynein-BicD complex to the plasma membrane (see Material and methods). The sorting activity of dynein was tracked by imaging mCherry-CAMSAP3. (H) Still images from time-lapses of a S2 cell before (left panel) and after addition of rapalog (middle and right panels). Magenta and green arrowheads represent the position of the membrane and the CAMSAP3, respectively. Note that the CAMSAP3 signal moves toward the cell body when rapalog is added while there is not a substantial retraction of the processes (see *Video 10*). If the same experiment is performed in DHC RNAi cells, addition of rapalog does not induce retrograde transport of the minus-end-out microtubules from the tips of processes (see *Video 10*). Scale bars, 10 µm. (I) Schematic representation of the BicD-dynein recruitment assays (microtubule minus-ends are represented in magenta). In the presence of LatB, dynein is soluble in the cytoplasm. Addition of rapalog directly recruits dynein to the membrane in the presence of BicD recruitment proteins, activating dynein's sorting activity and retrograde transport of microtubules to the cell body. (J) Fraction of processes that displayed retrograde movement of minus-end-out microtubules as imaged by CAMSAP3 signal. (Control, n=122 processes; DHC RNAi, n=99 processes). DHC, dynein heavy chain.

The following figure supplement is available for figure 3:

**Figure supplement 1.** Validation of the rapalog recruitment assays in S2 cells.

processes. In control cells, GFP-CAMSAP3 has a broad, scattered dot distribution (*Figure 2H*, left panel). Dynein depletion induced a striking *en masse* accumulation of CAMSAP3 at the tips of processes (*Figure 2H,I*). Overexposed images of those processes revealed that minus-ends could be still found in other areas of the processes (*Figure 2—figure supplement 2*). Identical phenotypes (mixed polarity of GFP-EB1 comets and accumulation of CAMSAP3 at the tips of the processes) were observed after knocking down several dynein cofactors (p150[Glued], Lis1 or NudE; *Figure 2G–I*; *Figure 2—figure supplement 1B* for knockdown efficiency). The *en masse* accumulation of microtubule minus-ends at neurite tips and S2 processes in dynein RNAi cells is consistent with the idea while kinesin-driven sliding stays intact, the sorting mechanism is inactivated by dynein depletion.

## Dynein must be recruited to the actin cortex to sort microtubules

In many organisms, the organization and orientation of the mitotic spindle requires attachment of cytoplasmic dynein to the cellular cortex (*Laan et al., 2012*; *Kiyomitsu and Cheeseman, 2013*). We hypothesized that organization of microtubules in axons, like their organization in the mitotic spindle, requires anchoring of dynein to the cortical network of actin filaments. To test this idea, we depolymerized actin filaments in cultured *Drosophila* neurons by treatment with LatB and quantified microtubule orientation in axons. We found that increasing concentrations of LatB gradually reduced the amount of F-actin, as judged by the intensity of rhodamine-phalloidin staining (data not shown). In parallel with F-actin depolymerization, axons displayed an increased fraction of microtubules with plus-ends facing the cell body (*Figure 3A*). At a high concentration of LatB (10 µM), when virtually no F-actin can be detected, about half of all EB1 comets moved toward the cell body, demonstrating

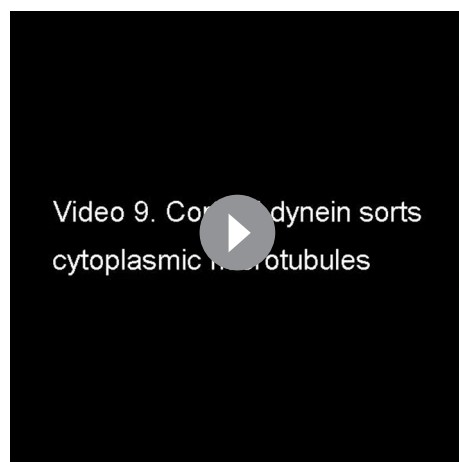

**Video 9.** Cortical dynein sorts cytoplasmic microtubules. Related to *Figure 3F*. Time-lapse images of S2 cells expressing GFP-CAMSAP3 cultured with 10 µM LatB. The drug was then washed out in control and DHC RNAi cells. Scale bars, 10 µm.

that depolymerization of actin induced random microtubule orientation in neurons (*Figure 3A*). Treatment with high LatB concentration also induced accumulation of microtubule minus-ends at the tips of the axons in *elav>GFP-CAMSAP3* neurons (*Figure 3B*).

These data show that actin depolymerization phenocopied dynein depletion in axons, suggesting that cortical dynein sorts microtubules. If this hypothesis is correct, LatB washout should restore dynein recruitment to the cortex, and the microtubule sorting activity of cortical dynein should remove minus-ends-out microtubules from the tip of the axon. To test this prediction, we washed out LatB and tracked the localization of GFP-CAMSAP3. We observed that the LatB washout induced removal of GFP-CAMSAP3 from axon tips without affecting the length of the axon (*Figure 3B*). It should be mentioned that LatB washout did not result in the normal scattered distribution of minus-ends along the length of the axon observed in control neurons (*Figure 2E*, left panels). Instead, they remained clustered in the shaft, most likely because by the time of drug washout these minus-end-out microtubules were already crosslinked into a bundle.

To observe the sorting activity of cortical dynein in real time, we again used S2 cells. As in the case of neurons, the treatment of S2 cells with high concentration of LatB induced formation of processes containing microtubules of mixed polarity (*Figure 3C*) and massive accumulation of minus-ends in their tips (*Figure 3D–F*; *Video 9*). Strikingly, LatB washout resulted in robust movement of GFP-CAMSAP3 labeled minus-ends from the tips of processes toward the cell body (*Figure 3F*, middle panel; *Figure 3G*; *Video 9*). However, if the same experiment was performed after dynein knockdown, retrograde transport of microtubule minus-ends was not observed and the minus-ends caps stay at the tips of processes (*Figure 3F*, right panel; *Video 9*), directly demonstrating the role of cortical dynein in removal of minus-end-out microtubules from processes to the cell body.

To further confirm that dynein and actin filaments are components of the same microtubule-sorting pathway, we decided to bypass the requirement for actin filaments by directly recruiting dynein to the cell membrane. For these experiments, we used a dynein recruitment tool developed by the Akhmanova and Hoogenraad labs (*Hoogenraad et al., 2003*; *Kapitein et al., 2010*), which contains the dynein activator Bicaudal D (BicD) fused to a FRB domain (FRB-GFP-BicD). The FRB-BicD-dynein complex can then be recruited to any region of interest by coexpression of a targeting protein coupled to a FKBP domain. This FKBP domain chemically dimerizes with the FRB domain in the presence of rapalog (a cell-permeable small molecule analog of rapamycin) (*Clackson et al., 1998*). To confirm that the FRB-FKBP dimerization system for dynein recruitment works in *Drosophila* S2 cells, we first coexpressed a GFP construct fused to FRB (FRB-GFP) and the peroxisome membrane-targeting signal peptide coupled to the red fluorescent protein (PEX3-RFP-FKBP). In the absence of rapalog, the GFP signal appeared soluble in the cytoplasm. The addition of rapalog recruited FRB-GFP to peroxisomes (*Figure 3—figure supplement 1A*). We next used this system to directly recruit dynein to the plasma membrane. For membrane targeting, we fused the transmembrane domain of GAP-43 (*Heim and Griesbeck, 2004*) with FKBP and coexpressed this construct with FRB-GFP-BicD in S2 cells. To visualize the recruitment of BicD to the membrane in these cells, we used total internal reflection fluorescence (TIRF) microscopy to image the GFP signal before and after addition of rapalog. Quantification showed that addition of rapalog significantly increased the intensity of the GFP fluorescence due to recruitment of cytoplasmic BicD to the plasma membrane (*Figure 3—figure supplement 1B*).

We next tested if direct recruitment of dynein to the membrane using rapalog could drive microtubule sorting in the absence of cortical actin. To test this hypothesis, mCherry-CAMSAP3 was used

to track the localization of microtubule minus-ends. In the absence of rapalog, CAMSAP3 accumulated in the processes of S2 cells treated with 10 µM LatB. Time-lapse imaging before addition of the drug revealed that these CAMSAP3 clusters, marking positions of minus-ends, were either static or pushing against the plasma membrane (*Video 10*). However, when the BicD-dynein complex was recruited to the membrane by addition of rapalog, CAMSAP3-labeled microtubule minus-ends robustly moved away from the tip, toward the cell body (*Figure 3H*, top panels; *Figure 3I*; *Video 10*). The same assay performed in DHC knockdown cells did not show this removal of microtubule minus-ends (*Figure 3H*, bottom panels; *Video 10*). Taken together, these data confirmed that dynein is responsible for microtubule sorting, and this activity requires the attachment of the motor to the cortex mediated by actin filaments.

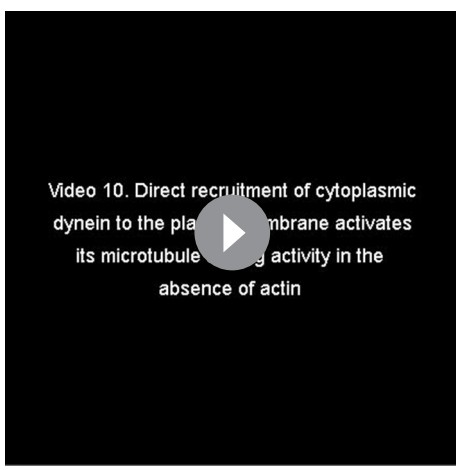

**Video 10.** Direct recruitment of cytoplasmic dynein to the plasma membrane activates its microtubule sorting activity in the absence of F-actin. Related to *Figure 3H*. S2 cells expressing GAP43-FKBP, FBP-GFP-BicD and mCherry-CAMSAP3 (control and DHC RNAi) were cultured in the presence of 10 µM LatB for 2 hr. To recruit BicD to the membrane 1 µM rapalog was added to the medium. Time-lapse videos were taken before and after addition of rapalog. Scale bar, 10 µm.

## Discussion

Mature axons contain microtubule arrays of uniform polarity with plus-ends facing away from the cell body (plus-end-out). Here, we tracked both plus-ends and minus-ends of microtubules in the axons of cultured *Drosophila* neurons at different stages of development. Our data showed that the axonal microtubule network undergoes dramatic reorganization during development. At early stages of neurite formation, the minus-ends of sliding microtubules are pushed against the plasma membrane, generating the forces that drive initial neurite outgrowth. At this stage, growing neurites contain microtubule arrays of mixed orientation. However, later in development, axonal microtubule arrays are reorganized from mixed to uniform (plus-end-out) orientation, consistent with the well established microtubule organization described in mammalian (*Baas et al., 1988*) and *Drosophila* axons (*Stone et al., 2008*). Here, we demonstrated that cortical dynein drives this transition by sliding and expelling minus-end-out microtubules from the axon toward the cell body.

### Kinesin-1 pushes the minus-ends of microtubules against the plasma membrane to initiate neurite formation

We previously demonstrated that initial neurite formation in *Drosophila* neurons requires microtubule-microtubule sliding driven by kinesin-1. Knockdown of kinesin-1 in primary neurons impairs the motility of interphase microtubules and causes neurons to fail to develop axons (*Lu et al., 2013b*). This was surprising because kinesin-1 is a plus-end microtubule motor and thus can only slide microtubules with minus-ends leading and plus-ends trailing. Furthermore, symmetry considerations dictate that kinesin slides antiparallel microtubules against each other; if microtubules are oriented parallel to each other kinesin would bundle rather than slide them (*Figure 1A*). Both of these considerations appear to contradict established literature about microtubule polarity in axons.

In this work, we performed two experiments that further support the idea that kinesin-1's sliding activity drives the initial stages of neurite formation (*Lu et al., 2013b*). First, nascent growing neurites contain microtubule arrays with mixed polarity. As mentioned above, this antiparallel orientation is required for microtubule sliding by kinesin-1 (*Figure 1A*). Second, microtubule minus-ends are pushed against the plasma membrane at the tips of processes, generating the force for neurite outgrowth.

## Role of other kinesins in axon formation

Recent experimental data revealed that 'mitotic' kinesins, such as kinesin-5 (*Nadar et al., 2008*; *Nadar et al., 2012*) and kinesin-6 (*Lin et al., 2012*; *del Castillo et al., 2015*), play an important role in the regulation of axon outgrowth. The well established function of these mitotic motors is to reorganize and stabilize the mitotic spindle to facilitate chromosome segregation. These motors accumulate in the spindle midzone where antiparallel microtubules coming from opposite poles overlap. Our work shows that developing *Drosophila* axons, like the spindle midzone, are filled with antiparallel microtubule arrays. This microtubule configuration suggests that the same mechanisms controlling the mitotic spindle could be used to regulate axonal outgrowth (*Baas, 1999*). Interestingly, 'mitotic kinesins' control neurite outgrowth both in mammalian and *Drosophila* systems, which supports the idea that the molecular players that drive axon initiation and outgrowth in *Drosophila* may be conserved across species.

## Cortical dynein sorts microtubules in developing axons

Our data show that minus-end-out microtubules are consistently observed at the early stages of neurite formation, while at later stages the vast majority of microtubules in axons have their plus-ends out, as have been reported in multiple published papers (*Baas et al., 1988*; *Stepanova et al., 2003*; *Stone et al., 2008*). Therefore, neurons must activate an additional sorting mechanism that eliminates these 'wrong' polarity microtubules from axons. Previous studies reported that dynein and dynein-associated proteins are required for correct microtubule orientation in axons of *Drosophila* da sensory neurons (*Zheng et al., 2008*; *Arthur et al., 2015*). Our results using cultured neurons are in full agreement with these studies, as dynein knockdown caused mixed microtubule polarity in axons. Interestingly, several studies have linked dynein-mediated microtubule reorganization to actin, both in interphase (*Mazel et al., 2014*) and in mitosis (*Kotak et al., 2012*; *Kiyomitsu and Cheeseman, 2013*). In this study, we showed that depolymerization of actin filaments (including cortical actin) using high concentrations of LatB causes the same defects in microtubule orientation as dynein knockdown. Recovery of cortical actin after drug washout results in efficient dynein-driven removal of minus-end-out microtubules from the tips of processes. Furthermore, the requirement for actin can be efficiently bypassed by direct recruitment of the dynein machinery directly to the plasma membrane. Therefore, we propose that microtubule sorting depends on the activity of cytoplasmic dynein attached to the cortical actin filament meshwork. However, as we did not get complete recovery of the wild type distribution of microtubules in our LatB washout or dynein-membrane recruitment experiments, it is likely that additional proteins that link actin and microtubule pathways in the neuron are involved in microtubule organization; one good candidate is the well characterized actin-microtubule crosslinking protein Short stop (*Lee and Kolodziej, 2002*).

## Additional roles of cortical dynein for microtubule organization in axons

In addition to the sorting activity described in this work, other groups have observed that dynein activity is required for axon elongation (*Ahmad and Baas, 1995*; *Ahmad et al., 1998*; *Grabham et al., 2007*). A more recent study by Miller et al. has reported that dynein generates forces that push the cytoskeletal meshwork forward during axonal elongation in cultured chick sensory neurons (*Roossien et al., 2014*). In agreement with their data, we predict that cortical dynein can promote axon outgrowth after the initial stages of neurite outgrowth, when microtubules are sorted into a plus-end-out orientation. Because the direction of dynein-powered forces is dictated by the intrinsic orientation of microtubules, microtubules with minus-end-out are redirected toward the cell body, whereas plus-end-out microtubules are transported toward the tip of the axon. Therefore, dynein-driven microtubule transport can not only remove microtubules with the wrong orientation (minus-end-out) but also push microtubules of the right orientation (plus-end-out) toward the axon tip.

In addition to the role of dynein in developing neurons, dynein may play an important role in axonal microtubule maintenance. Microtubule polymers, like other protein structures, require subunit turnover to maintain their integrity. It has been proposed that several microtubule severing proteins are able to fragment microtubules into short pieces (*McNally and Vale, 1993*). In this scenario, short microtubule seeds are generated which may be reoriented by diffusion forces in either direction (plus-end-in or plus-end-out) with equal probability. Indeed, active bidirectional transport of short

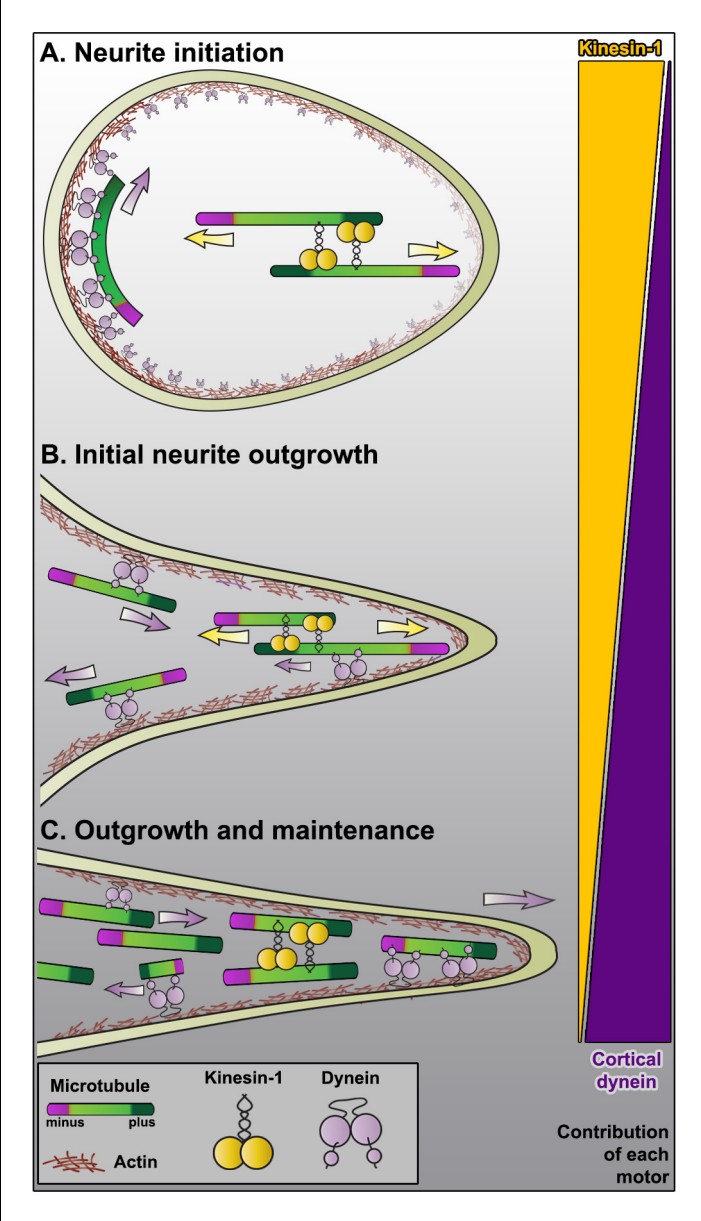

**Figure 4.** Model of microtubule sliding and axon formation. (**A**) Kinesin-1 induced sliding of antiparallel microtubules initiates formation of processes. Note that at this stage cortical dynein can only slide microtubules parallel to the plasma membrane, suggesting that dynein is not involved in the initiation of processes. (**B**) Short neurites contain antiparallel microtubule arrays. Under this configuration, kinesin-1 still slides microtubules with their minus-ends out, and cortical dynein can start removing minus-end-out microtubules to the cell body. (**C**) Due to continuous dynein-powered sorting activity, the growing axon is mostly filled with uniformly oriented plus-end-out microtubules. At this stage, dynein can continue removing minus-end-out microtubules, and can contribute to axon elongation by pushing plus-end-out microtubules toward the tip (see *Roossien et al., 2014*). Note that at this stage, kinesin-1 motors only bundle parallel microtubules and no longer contribute to microtubule sliding or outgrowth. The triangles on the right represent the contribution of the motors at each stage.

microtubule fragments in the axons has been described in cultured rat hippocampal neurons (*Liu et al., 2010*). The authors proposed that the transport of these short microtubule fragments is driven by dynein. Our prediction is that microtubule fragments transported in the retrograde direction should be oriented with plus-ends toward the cell body while anterogradely transported fragments should have the opposite orientation.

## A unified model of microtubule sliding and axon formation

It has been shown in many different model systems that kinesin-1 and dynein are the two major motors involved in reorganizing cytoplasmic microtubules (*Fink and Steinberg, 2006*; *Straube et al., 2006*; *Jolly et al., 2010*; *Mazel et al., 2014*). Both motors play an important role in axon formation. However, the contribution of each motor likely differs at different developing stages (see model described in *Figure 4*). In spherically shaped undifferentiated cells, kinesin-1 motors slide antiparallel microtubules perpendicularly to the plasma membrane with their minus-end-leading (*Figure 4A*). This sliding generates the force that breaks the symmetry of the cell and induces a deformation of the plasma membrane to initiate neurite outgrowth. During this stage, the role of cortical dynein is probably limited by steric restrictions. Cortical dynein can only slide microtubules tangential to the plasma membrane, and therefore dynein-mediated movement cannot deform the membrane and initiate outgrowth. At the next stage, nascent neurites contain antiparallel microtubule arrays (*Figure 4B*). This configuration allows kinesin-1 to continue to drive antiparallel microtubule sliding. Once the processes start to form, the new geometry will allow cortical dynein to contribute to both microtubule organization and neurite outgrowth. Cortical dynein in the neurite cortex can only slide microtubules parallel to the axis of the process. This contribution of cortical dynein likely increases as the processes grow longer (engaging more cortical dynein molecules) and thinner (allowing dynein to reach a larger fraction of microtubules in the processes).

There are two possible microtubule orientations in the neurite that can interact with cytoplasmic dynein. Microtubules with minus-end-out (microtubule of the 'wrong' polarity) will be moved by dynein toward the cell body, therefore eliminating them from the processes. At the same time, dynein can interact with and push plus-end-out microtubules moving them toward axon tips, and thus contributing to the forces that drive axonal growth (*Figure 4B*). As a result of this continuous sorting activity of dynein, developed axons contain uniform microtubule arrays with their plus-end distal (*Figure 4C*). Under this new microtubule configuration, kinesin-1 contributes little to microtubule sliding (as antiparallel microtubule arrays are needed for kinesin to engage in sliding), but instead favors its bundling activity (*Figure 1A*, right panel). In addition to its contribution to neurite outgrowth, cortical dynein may have an important role in axon maintenance, removing nascent microtubules of wrong orientation that may appear in the axon either due to new microtubule polymerization or due to the severing of pre-existing microtubules. Because axons and dendrites are common morphologic features observed in neurons from ancestral metazoans to mammals, we postulate that the same molecular mechanism presented here for axon formation in *Drosophila* neurons is conserved in other organisms. Of course, this 'microtubule-centric' model does not describe all the mechanisms involved in axon formation, and clearly leaves out the critical question of axon guidance, but it provides clear roles for multiple microtubule motors involved in axon formation and microtubule organization.

## Cortical dynein may dictate neurite identity

Microtubule orientation differs between axons and dendrites. While axons are filled with uniformly oriented microtubules with plus-end-out, dendrites contain either microtubules of mixed orientation (mammalian neurons) (*Baas et al., 1988*) or a majority of microtubules minus-end-out (*Drosophila* and *C. elegans* neurons) (*Stone et al., 2008*; *Yan et al., 2013*). We speculate that neurons selectively employ cortical dynein to dictate which of the nascent neurites will become the future axon. It is likely that the microtubule sliding activity or efficiency of dynein recruitment may be downregulated in dendrites. This hypothesis is in agreement with a recent study in *Drosophila* da sensory neurons showing that disruption of NudE, a dynein cofactor, impairs microtubule orientation in axons, without affecting the orientation of dendritic microtubules (*Arthur et al., 2015*). Furthermore, Yan et al. directly demonstrated that kinesin-1 (*unc-116*) is required for minus-end-out orientation of microtubules in dendrites (*Yan et al., 2013*). Those observations support the idea that the activity of cortical dynein is downregulated in dendrites, thus preserving the initial minus-end-out orientation of microtubules created by kinesin. Future experiments are required to unravel how dynein interacts with cortical actin and how dynein's microtubule-sorting activity is regulated in neurons.

## Material and methods

### Plasmids and cloning

To visualize microtubule minus-ends in the *Drosophila* S2 cell system, a cDNA encoding mouse CAMSAP3 (*Jiang et al., 2014*) was cloned into the pMT-GFP and pMT-mCherry backbones by NotI-AgeI restriction enzyme sites to generate GFP-CAMSAP3 and mCherry-CAMSAP3. GFP-CAMSAP3 was also cloned into UASp backbone by KpnI-XbaI to create a transgenic *Drosophila* fly line containing UASp-GFP-CAMSAP3 by standard P element-mediated transformation. A plasmid encoding EB1-GFP under endogenous EB1 promoter (pMT-EB1:EB1-GFP) was used to visualize microtubule plus-ends in S2 cells. For recruitment experiments, all constructs were cloned into pAC.V2014, a modified version of pAC5.1 containing the following multiple cloning site (KpnI-NheI-BmtI-HindIII-AscI-EcoRI-NotI-XbaII-EcoRV-XhoI). FRB-GFP and PEX3-mRFP-FKBP fragments from pβActin-GFP-FRB and pβactin-PEX3-mRFP-FKBP plasmids (*Kapitein et al., 2010*) were subcloned in the HindIII-NotI and HindIII-EcoRI sites to generate pAC.V2014-FRB-GFP and pAC.V2014-PEX3-mRFP-FKBP. *Drosophila* BicD cDNA (LD17129 from DGRC) was amplified by PCR and ligated in the AscI-NotI sites of pAC.V2014-FRB-GFP to create pAC.V2014-FRB-GFP-BicD. FKBP fragment was inserted in pAC.V2014 using the AscI-EcoRI sites to create pAC.2014-FKBP. To recruit the FKBP domain to the membrane, the DNA sequence encoding the transmembrane domain GAP-43 (MLCCMRRTKQVEK-NDEDQKI) was ligated in the KpnI-AscI sites of pAC2014-FKBP to create pAC2014-GAP43-FKBP.

### Fly stocks and genetics

Fly stocks and crosses were cultured on standard cornmeal food based on Bloomington Stock Center's recipe at room temperature. The following fly stock lines were used in this study: *UASp-tdEOS2-αtub84B* (2nd and 3rd chromosome insertions) (*Lu et al., 2013b*), *ubi-EB1-GFP* (3rd chromosome insertion) (*Shimada et al., 2006*); *UASt-EB1-GFP* (3rd chromosome insertion) (*Rolls et al., 2007*); *elav-Gal4* (3rd chromosome insertion, Bloomington stock #8760) (*Luo et al., 1994*); *UASp-GFP-CAMSAP3* (2nd chromosome insertion, created in this study); DHC64C-RNAi TRiP lines, (Valium 20, Bloomington stock #36698, 3rd chromosome attP2 insertion, targeting DHC64C CDS 1302–1322; Valium 22, Bloomington stock #36583, 2nd chromosome attP40 insertion, targeting DHC64C CDS 10044–10064). Stocks of yw; *DHC64C-TRiP RNAi-Valium22*; *ubi-EB1-GFP*,yw; *UASt-EB1-GFP*; *DHC64C-TRiP RNAi-Valium20, and yw*; *UASp-GFP-CAMSAP3*; *DHC64C-TRiP RNAi-Valium20* were generated using standard balancing procedures, and crossed with *elav-Gal4* to examine EB1-GFP or GFP-CAMSAP3 in DHC64C knockdown.

### *Drosophila* cell culture: primary neurons and S2 cells

Primary neurons were obtained from brains of 3rd instar larva as previously described (*Lu et al., 2015*). Neurons were plated onto Concanavalin A-coated coverslips in supplemented Schneider's medium (20% fetal bovine serum, 5 µg/ml insulin, 100 µg/ml penicillin, 100 µg/ml streptomycin, and 10 µg/ml tetracycline). For actin depolymerization assays, neurons were plated in Xpress medium for 2 hr before addition of LatB. *Drosophila* S2 cells were cultured as previously described (*Barlan et al., 2013*). To induce the formation of processes in S2 cultures, cells were plated in the presence of 0.5 µM LatB. For knockdown assays in S2 cells, cultures at $1.5 \times 10^6$ cells/mL were treated twice with 20 µg of dsRNA (day 1 and day 3) and cell analysis was performed on day 5. Double-stranded RNA was transcribed in vitro with T7 polymerase, and purified using LiCl extraction. Primers used to create T7 templates from fly genomic DNA were as follows. T7 promoter sequences (TAATACGACTCACTATAGGG) were added to the 5′ end of each primer). DHC, forward, AAACTC-AACAGAATTAACGCCC; reverse, TTGGTACTTGTCACACCACT (*Jolly et al., 2010*); p150[Glued], forward GAGTTTGAGGAGACGATGGACCACC, reverse GTTGCACGATGGGGTTTCCTTTGCAG; Lis1, forward GGTTGAATTACGCGATCATGAGCATACTGTGGA, reverse GGAGGTGCAGAAATGCTGAT-GCGCGTATAG; NudE, forward GCTCAAGTTGGAATCGCATGGCATCGATATGTC, reverse CTCT-CGTCTCATCCATTAATCGCTGTAGTTTTTCCTGC. To induce the formation of short microtubule fragments, *Drosophila* S2 cells stably expressing GFP-CAMSAP3 were treated with 25 µM Vinblastine for 1 hr. Soluble fraction of tubulin was removed using BRB80 buffer (80 mM PIPES buffer (pH 6.8), 1 mM EGTA, 1 mM DTT, and 1 mM $MgCl_2$) supplemented with 1% Triton X-100. Extracted cells were fixed and microtubules were immunostained with mouse anti-α-tubulin (DM1α).

## Microscopy and image acquisition

To image EB1 comets, CAMSAP3 localization and lysosome transport in *Drosophila* S2 cells and primary neurons, a Nikon Eclipse U2000 inverted microscope equipped with a Yokogawa CSU10 spinning disk head, Perfect Focus system (Nikon) and a 100 X 1.45 NA lens was used. Images were acquired using Evolve EMCCD (Photometrics) and controlled by Nikon Elements 4.00.07 software. For EB1 and CAMSAP3 time-lapses, images were collected every 2 s for 1 min (EB1 comets) and every 1 min or 5 min for 16 min or 60 min, respectively (CAMSAP3). To image the plasma membrane, CellMask Deep Red dye (1:10,000) was added in both cultured neurons and S2 cells 5 min before imaging. For To visualize sliding, a small fraction of microtubules in cultured neurons expressing *tdEOS-αTub84B* was photoconverted. Photoconversion was performed by confining the illumination of a heliophore laser (405 nm) in the epifluorescence pathway using a diaphragm. Images were collected once per 30 s for 5 min. To image the recruitment of BicD to the plasma membrane and the distribution of CAMSAP3 in mature neurons, TIRF images were collected using a Nikon Eclipse U2000 inverted microscope equipped with a Plan-Apo TIRF 100×/1.45 NA objective and a Hamamatsu CMOS Orca Flash 4.0 camera (Hamamatsu Photonics, Hamamatsu, Japan), controlled by MetaMorph 7.7.7.0 software (Molecular Devices, Downingtown, PA). Statistical significance for CAMSAP3 populations was determined using two-tailed Fischer's test with a confidence interval of 95%.

## Analysis of EB1 comets

The orientations of EB1 comets both in cultured neurons and S2 cells were quantified using MATLAB and ImageJ. EB1 comets were tracked using the plus-end tracking algorithm in u-track 2.0, developed by Gaudenz Danuser's group (*Jaqaman et al., 2008*; *Matov et al., 2010*). The x-y positions of EB1 comets were extracted and loaded into a custom ImageJ plugin. This semi-automated plugin defined the center of cells and determined the angle of each comet's trajectory as compared with the cell center. For curved and serpentine axons, trajectories were subdivided into linear segments before ImageJ analysis to ensure that comet orientation was correctly identified. Comet trajectories with an angle >290° or <70° compared with the cell center were defined as plus-end-out. Comet trajectories with an angle >110 and <250 degrees were defined as plus-end-in. Only comets contained in processes were included in the analysis. To create kymographs presented in *Figure 1E and Figure 2A* the Reslice plugin developed in FIJI was used. Statistical significance for EB1 comets was determined using the non-parametric Mann-Whitney test with a confidence interval of 95%. This test analysis compares the distributions of two unmatched groups.

## Antibodies for Western blot

For Western blot analysis of S2 cell extracts, the following primary antibodies were used: anti-DHC monoclonal antibody 2C11-2 (*Sharp et al., 2000*) and rabbit polyclonal antibody against KHC head domain provided by A. Minin (Institute of Protein Research, Russian Academy of Sciences, Moscow, Russia).

## Recruitment experiments

The formation of the FRB-FKBP complex in all recruitment experiments was induced by addition of 1 μM rapalog (final concentration) (A/C Heterodimerizer, Clontech) to the culture medium. To test the efficiency of recruitment experiments, *Drosophila* S2 cells were transiently cotransfected either with plasmids encoding FRB-GFP and PEX3-mRFP-FKBP or FRB-GFP-BicD and GAP43-FKBP. Cells were imaged before and 30 min after addition of rapalog to the medium. To directly recruit endogenous dynein to the membrane, S2 cells were cotransfected with plasmids codifying GAP43-FKBP, FRB-GFP-BicD and mCherry-CAMSAP3 in the following ratio (3:1:1). Cells were plated with 10 μM LatB for 2 hr to allow the accumulation of microtubule minus-ends at process tips. Distribution of mCherry-CAMSAP3 was tracked by time-lapse imaging before and after addition of the drug (to the final concentration of 1 μM).

## Acknowledgements

The authors thank the Bloomington Stock Center (NIH P40OD018537) for fly stocks, and *Drosophila* Genomics Research Center (NIH 2P40OD010949-10A1), S Rogers (UNC Chapel Hill), A Akhmanova and C Hoogenraad (both from Department of Biology, Utrecht University) for plasmids. Research reported in this publication was supported by the National Institute of General Medical Science of the National Institutes of Health under award number R01GM052111.

## Additional information

### Funding

| Funder | Grant reference number | Author |
| --- | --- | --- |
| National Institute of General Medical Sciences | GM 052111 | Vladimir I Gelfand |

The funders had no role in study design, data collection and interpretation, or the decision to submit the work for publication.

### Author contributions

UDC, Conception and design, Acquisition of data, Analysis and interpretation of data, Drafting or revising the article; MW, Acquisition of data, Analysis and interpretation of data; WL, Acquisition of data, Drafting or revising the article; VIG, Conception and design, Analysis and interpretation of data, Drafting or revising the article

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
