## [Decision Letter]

Thank you for submitting your work entitled "Interplay between kinesin and cortical dynein during axonal outgrowth and microtubule organization in *Drosophila* neurons" for peer review at *eLife*. Your submission has been favorably evaluated by Vivek Malhotra (Senior editor) and three reviewers, one of whom is a guest Reviewing Editor.

The reviewers have discussed the reviews with one another and the Reviewing editor has drafted this decision to help you prepare a revised submission.

Summary:

The reviewers find that this interesting study provides direct support for sequential roles of kinesin-1 and dynein in extending axons with the typical plus-ends-out microtubule orientation.

Principal conclusions:

1) Kinesin-1 drives neurite/process extension by pushing minus-end-out microtubules against the cell membrane. This is supported by live cell imaging using a GFP-CAMSAP3 as a marker for microtubule minus ends in cultured *Drosophila* neurons and S2 cells.

2) After the initial extension phase, dynein at the cell cortex then removes the minus-end-out microtubules by sliding them back into the cell body, while the plus-end-out microtubules remain.

3) An interaction with the actin network recruits dynein to the cell cortex.

Although previous work has suggested a role for dynein in generating microtubule orientation in neurons, this study tests that hypothesis directly, as well as expanding on the mechanism by which kinesin-1 drives process extension.

Essential revisions:

The reviewers raise a number of concerns that must be adequately addressed before the paper can be accepted. Some of the required revisions will require further experimentation within the framework of the presented studies and techniques.

1) The study rests on the use of GFP-CAMSAP3 as a marker for microtubule minus ends, but the evidence for this conclusion needs to be strengthened substantially. Imaging EB1-GFP during the early stages of process extension would be a simple assay to test whether CAMSAP3 is indeed present only on minus ends. The use of EB1-GFP would also demonstrate that plus-end-out microtubules do not drive process extension: a claim in the manuscript for which there is no direct proof at the moment. Scoring of microtubule orientation versus process growth is also needed, rather than showing a single example.

2) Related to point 1, some of the data seems to directly contradict the conclusions of the paper, and this needs to be clarified. For example, quantitation of EB1 orientation reveals that cells depleted of DHC have a 50:50 mix of microtubule polarity (Figure 2 and Figure 3), and yet CAMSAP3 is strongly accumulated at process tips (Figure 2 and Figure 3). Is this because of timing? Furthermore, Figure 2 shows all microtubules growing +ends out, but 2G (control) shows minus ends at the process tips.

3) A major conclusion of the work is that cortical dynein drives sliding of the 'wrong' orientation microtubules out of older processes, but currently there is limited direct evidence for that. For example, although we see accumulation of GFP-CAMSAP3 in S2 cells depleted of DHC, only EB1-GFP orientation (not CAMSAP3) in older neurons is shown. Is the same striking phenotype observed there? Movies showing dynein-driven sliding in older axons should be provided.

In addition, the data showing that rapalog-stimulated recruitment of dynein to the membrane drives reorientation of microtubules needs strengthening, since the cell process shown in Figure 3 and Video 9 appears to be retracting. Can better proof of sliding be shown? Would photoactivation of CAMSAP3 be helpful? Furthermore, this important experiment needs to be quantitated.

4) Based on the effects of actin depolymerization, the authors conclude that dynein is recruited to the cell membrane/cortex via interaction with actin. This is quite different to the previously reported mechanisms of dynein association with the plasma membrane. It also ignores possible contributions of proteins that might link actin and microtubules, such as Shot. In the absence of any direct proof, this claim should be toned down.

---

## [Author Response]

Essential revisions:

*The reviewers raise a number of concerns that must be adequately addressed before the paper can be accepted. Some of the required revisions will require further experimentation within the framework of the presented studies and techniques. 1) The study rests on the use of GFP-CAMSAP3 as a marker for microtubule minus ends, but the evidence for this conclusion needs to be strengthened substantially. Imaging EB1-GFP during the early stages of process extension would be a simple assay to test whether CAMSAP3 is indeed present only on minus ends. The use of EB1-GFP would also demonstrate that plus-end-out microtubules do not drive process extension: a claim in the manuscript for which there is no direct proof at the moment. Scoring of microtubule orientation versus process growth is also needed, rather than showing a single example.*

The location of proteins of the CAMSAP family on microtubules has been extensively characterized in the last few years in several systems (Tanaka et al., 2012; Hendershott and Vale, 2014; Jiang et al., 2014; Yau et al., 2014). These publications have demonstrated that CAMSAPs label minus-ends of microtubules exclusively and play an important physiological role in minus-end stabilization. Our data show that ectopically expressed mammalian CAMSAP3 tagged with GFP labeled only one end of microtubules in *Drosophila* cells, and live imaging of these cells demonstrated that labeled ends were stable, in agreement with the published data.

To further validate our conclusion that CAMSAP3 labels only minus-ends in *Drosophila*, as requested by the reviewers, we performed an additional experiment in S2 cells. We coexpressed EB1-GFP and mCherry-CAMSAP3 (see new Figure 1—figure supplement 1 and new Video 3). Colocalization of EB1 comets and CAMSAP3 signal was never observed, and as can be seen in Video 3, EB1 and CAMSAP3 label opposite ends of the same microtubule. This further validates the use of CAMSAP3 as a minus-end marker.

The reviewers pointed out the potential role of microtubule plus-ends in process formation. This is an important question that we did not fully address in our original manuscript. To explore this possibility, we performed new experiments to test whether EB1 comets localize at the tips of growing processes in S2 cells. Briefly, in actively growing processes of S2 cells, we simultaneously imaged microtubule plus-ends with EB1-GFP and process tips outlined with a plasma membrane dye. We have shown a representative example of this experiment in the new Figure 1. Note that during the initial outgrowth of processes, EB1 does not colocalize with process tips. Furthermore, as the reviewers requested, we quantified the colocalization of plus- or minus-end markers with the tips of growing processes seen with a membrane dye (see new Figure 1). We found that minus-ends colocalized with process tips during 94% of outgrowth events (n=55 processes). However, plus-ends only localized to the tips of processes during 33% of outgrowth events (n=51 processes). Based on this data, we believe that microtubule minus-ends play the critical role in the early stages of outgrowth, but it is possible that plus ends have some contribution to the growth, especially at the later stages (subsection “Microtubule minus-ends push neurite tips at the initial stages of process formation”, fifth paragraph and subsection “Additional roles of cortical dynein for microtubule organization in axons“, first paragraph).

*2) Related to point 1, some of the data seems to directly contradict the conclusions of the paper, and this needs to be clarified. For example, quantitation of EB1 orientation reveals that cells depleted of DHC have a 50:50 mix of microtubule polarity (Figure 2 and Figure 3), and yet CAMSAP3 is strongly accumulated at process tips (Figure 2 and Figure 3). Is this because of timing?*

The reviewers pointed out an apparent contradiction between data obtained with EB1 and CAMSAP3 markers in DHC RNAi processes: EB1 comets demonstrate a mixed polarity of microtubules in processes, while process tips have an accumulation of minus ends. In fact, we see no contradiction here, as the cartoon (left) and the confocal image (right) (S2 cell/GFP-CAMSAP3) below help to explain. Accumulation of minus ends at process tips only means that the most distal microtubules have their minus-end at the tips; these microtubules are deposited to the tip by kinesin-driven sliding. However, microtubules located closer to the cell body can (and do) have mixed polarity. The location of minus ends in the shafts of the processes may not be obvious in Figure 3 because the signal from individual GFP-CAMSAP3 minus ends within these regions is much dimmer that of the many clustered minus-ends at the tip. In fact, adjustment of contrast of the boxed area in the right panel clearly shows multiple GFP-CAMSAP3 speckles in the shaft (bottom right panel), as predicted by the cartoon on the left. To avoid readers’ confusion, we have clarified this point in the text (subsection “Dynein sorts microtubules in *Drosophila* axons“, last paragraph).

*Furthermore, Figure 2 shows all microtubules growing +ends out, but 2G (control) shows minus ends at the process tips.*

We agree with the reviewers that our original Figure 2 (now Figure 2) was confusing for the reader. This confusion might have arisen because the original figure depicted the orientation of comets from the first frame; we did not show the direction of new comets that originated later in the sequence. Quantification of all EB1 comets throughout a time series for each cell revealed that processes in S2 cells contained around 80% plus-end-out and 20% plus-end-in comets (see new Figure 2). Based on these results, it is expected that a small fraction of microtubule minus-ends might be present at the tips of processes. We modified Figure 2 by adding new arrows to show directions of comets initiated later in the sequence. All comets from this specific example can be easily seen and tracked in our Video 8.

*3) A major conclusion of the work is that cortical dynein drives sliding of the 'wrong' orientation microtubules out of older processes, but currently there is limited direct evidence for that. For example, although we see accumulation of GFP-CAMSAP3 in S2 cells depleted of DHC, only EB1-GFP orientation (not CAMSAP3) in older neurons is shown. Is the same striking phenotype observed there? Movies showing dynein-driven sliding in older axons should be provided.*

We took this reviewer’s comment very seriously and generated a fly expressing GFP-CAMSAP3 and DHC shRNA driven by the GAL4 promoter. These flies were crossed with elav>Gal4 to induce expression of both transgenes in neurons. The third instar larvae (*elav>GFP-CAMSAP3 + DHCi*) developed the same survival and locomotion defects than were observed in elav>DHC RNAi alone, indicating that the DHC shRNA efficiently depleted DHC even when Gal4 drove GFP-CAMSAP3 and DHC shRNA simultaneously. We incorporated a new Figure 2 to show the localization of GFP-CAMSAP3 in *elav>GFP-CAMSAP3 and elav>GFP-CAMSAP3 + DHCi*. Note that depletion of DHC induced accumulation of GFP-CAMSAP3 in the tips of neurites, just as it did after DHC depletion in S2 cell processes, directly answering the reviewer’s question.

To show dynein-driven sliding in axons, we first treated neurons with a high concentration of LatB (10 µM) for 48 hr. This induced accumulation of GFP-CAMSAP3 in neurite tips (new Figure 3, left panel). To activate microtubule-microtubule sorting, we washed out LatB and followed the localization of the CAMSAP3 signal and membrane tip. As we show in the new Figure 3 (right panels) LatB washout caused the removal of minus-ends from neurite tips without affecting the length of the axon. We have to point out, however, that the minus-end clusters do not get expelled all the way to the cell body, most likely because at this stage, axons contain highly bundled minus-end-out microtubules. See the new data in subsection “Dynein must be recruited to the actin cortex to sort microtubules”, second paragraph.

*In addition, the data showing that rapalog-stimulated recruitment of dynein to the membrane drives reorientation of microtubules needs strengthening, since the cell process shown in Figure 3 and Video 9 appears to be retracting. Can better proof of sliding be shown? Would photoactivation of CAMSAP3 be helpful? Furthermore, this important experiment needs to be quantitated.*

To address this important point, we repeated and improved these experiments (see the new Figure 3 and new Video 10). The new data show a better example of dynein sorting activity after addition of rapalog and no process retraction. Furthermore, we added a new experimental condition (DHC RNAi), which demonstrates that dynein depletion prevents the removal of minus-end-out microtubules after addition of rapalog. As reviewers requested, we quantified the removal of minus-end clusters from the tips after the addition of rapalog. The quantification shows that, in control cells, minus-end clusters were removed from about 70% of tips after addition of rapalog, while in DHC RNAi-treated cells, removal was observed in only 3% (new Figure 3).

*4) Based on the effects of actin depolymerization, the authors conclude that dynein is recruited to the cell membrane/cortex via interaction with actin. This is quite different to the previously reported mechanisms of dynein association with the plasma membrane. It also ignores possible contributions of proteins that might link actin and microtubules, such as Shot. In the absence of any direct proof, this claim should be toned down.*

We completely agree with the reviewer’s comment. Our results experimentally demonstrate that cortical dynein is involved in sorting microtubules of *Drosophila* processes. Our experiments place dynein activity downstream of actin in the microtubule organization pathway. However, at this moment it is unknown whether dynein interacts with actin directly or through adaptors (as, for example, has been extensively demonstrated in case of spindle orientation in dividing cells). Furthermore, as correctly pointed out by the reviewer, actin can be involved in microtubule organization via additional mechanisms (such as actin-microtubule cross-linker short stop). Therefore, we have toned down the direct interaction hypothesis in the manuscript. In addition, we mentioned in the Discussion section, the potential contribution of proteins known to link microtubules (subsection “Cortical dynein sorts microtubules in developing axons”).